# Health impacts of wildfire-related air pollution in Brazil: a nationwide study of more than 2 million hospital admissions between 2008 and 2018

Weeberb J. Requia [1✉], Heresh Amini [2,3], Rajarshi Mukherjee[4], Diane R. Gold[5,6] & Joel D. Schwartz [7]

We quantified the impacts of wildfire-related $PM_{2.5}$ on 2 million hospital admissions records due to cardiorespiratory diseases in Brazil between 2008 and 2018. The national analysis shows that wildfire waves are associated with an increase of 23% (95%CI: 12%–33%) in respiratory hospital admissions and an increase of 21% (95%CI: 8%–35%) in circulatory hospital admissions. In the North (where most of the Amazon region is located), we estimate an increase of 38% (95%CI: 30%–47%) in respiratory hospital admissions and 27% (95%CI: 15%–39%) in circulatory hospital admissions. Here we report epidemiological evidence that air pollution emitted by wildfires is significantly associated with a higher risk of cardiorespiratory hospital admissions.

[1] School of Public Policy and Government, Fundação Getúlio Vargas Brasília, Distrito Federal, Brazil. [2] Department of Public Health, University of Copenhagen, Copenhagen, Denmark. [3] Harvard T.H. Chan School of Public Health, Harvard University, Boston, Massachusetts, USA. [4] Department of Biostatistics, Harvard T.H. Chan School of Public Health Boston, Boston, Massachusetts, USA. [5] Harvard T.H. Chan School of Public Health, Harvard University Boston, Boston, Massachusetts, USA. [6] Channing Division of Network Medicine, Department of Medicine, Brigham and Women's Hospital and Harvard Medical School, Boston, Massachusetts, USA. [7] Department of Environmental Health, Harvard TH Chan School of Public Health Boston, Boston, Massachusetts, USA. ✉email: weeberb.requia@fgv.br

Wildfires have burned a large number of areas in the past years. Mouillot and Field[1] estimate that each year about 6 million km² of vegetation is burned globally. There are wide differences in fires around the world. Africa is very fire-prone, where about 50% of West African savannas are burned each year[2,3]. In Europe, ~18,000 km² are burned annually[4]. In the United States, Koplitz et al.[5] estimate 90,000 km² of burned area per year. In Brazil, according to the National Institute of Spatial Research–INPE (http://queimadas.dgi.inpe.br/queimadas/), between January 2020 and August 2020, there were about 120,000 km² of burnt area.

Forest fires cause many environmental impacts including air pollution[6]. Wildfires emit substantial amounts of air pollutants that can travel over large distances, affecting air quality and human health far from the originating fires[4]. Fine particulate matter ($PM_{2.5}$) is the major pollutant emitted by wildfires. In the United States, according to the National Emissions Inventory, in 2014 wildfires represented more than 20% of total $PM_{2.5}$ emissions annually[7]. About 12–16% of global wildfire-related particulate emissions occur across Brazil[8]. Tropospheric ozone ($O_3$) is another pollutant related to wildfires. In Brazil, biomass burning is a substantial fraction of the precursors for $O_3$ formations, where wildfires contribute between 23% and 41% of the total $O_3$ during the pollution events[9].

Regarding the health impacts of wildfire smoke exposure, a critical review of 53 epidemiological studies[10] shows that wildfire exposure is associated with respiratory diseases and growing evidence suggests associations for specific cardiovascular endpoints. The literature consists of studies for both hospital admissions and mortality as a health outcome. For example, Liu et al.[11] estimated a 7.2% increase in risk for respiratory admissions during smoke wave days with high wildfire-specific $PM_{2.5}$ during 2004 and 2009 in the Western United States. In Brazil, the reduction of wildfire-related particulate emissions by 30% indicates a health improvement by preventing about 400 to 1700 premature adult deaths annually[8].

Most of the previous investigations on the population health effects from wildfire exposure were performed in the United States[11–13] and Australia[14,15]. Little research has focused on Brazil. We searched in PubMed and Web of Science using the following keywords: wildfire, air pollution, health effects, human exposure, and Brazil. We found only four studies[8,16–18] and three of them focused only on the Amazon region. More research is needed to explore the nationwide effect of wildfires on health, to clarify whether health outcomes are associated with wildfire smoke, to explore the effect of spatiotemporal factors and to investigate whether certain populations are more susceptible.

Given that Brazil is very fire-prone region and (i) where there are different types of biomass (Amazon Forest, Cerrado, Atlantic Forest, Caatinga, Pampa, and Pantanal) that are strongly correlated with wildfire events over space and time—e.g., according to the INPE, Caatinga region is projected to become warmer and dryer with potentially large impacts on dust and wildfire; (ii) where there is a critical challenge related to land use (e.g., agriculture, deforestation etc.), which also correlates to wildfire occurrence—e.g., from 1990 to 2011, the land used for cropping in Brazil grew from ~530,000 to ~680,000 km² and 60% to 80% of deforested land are related to pastures for beef production[19]; and (iii) where there is a considerable difference in the quality of health/environment and healthcare across different populations (influencing health/environment equity in negative ways), which is an important determinant of the health impacts of wildfire-related air pollution[20]; further epidemiological studies in Brazil are essential, which could provide a better support for policy-makers with the objective of improving the quality of public health. Our research addresses this gap by quantifying the impacts of wildfire-related $PM_{2.5}$ on hospital admissions due to cardiorespiratory diseases in Brazil between 2008 and 2018. In particular, we investigated this impact by using a modeled approach that defines wildfire waves accounting for spatio-temporal heterogeneity on a population sample of more than 2 million patients.

## Results

**Hospital admission characteristics**. Our study population includes 2,044,038 hospital admissions for cardiorespiratory diseases in Brazil between 2008 and 2018. Among those, 50.2% are respiratory diseases and 49.8% circulatory diseases. Males were the majority of the patients for both respiratory and circulatory diseases. For the age group in the respiratory diseases, the largest proportion was for the patients aged <5 years (25%) and patients >64 years (27%). For the circulatory hospital admissions, the largest percentage was for the patients 35–64 years (46%) and patients >64 years (45%). Table 1 provides the descriptive characteristics of these health events and in Fig. 1 we show the nationwide occurrences of respiratory and circulatory hospital admissions in Brazil.

**Exposure characteristics**. Summary statistics for wildfire, air pollution, and meteorological variables are presented in Table 2. The mean concentration of $PM_{2.5}$ over the study period in Brazil was ~15 µg m$^{-3}$, which exceed the air quality guidelines from the World Health Organization (10 µg m$^{-3}$ annual mean). For wildfire records, we estimated a mean of 4.37 and a maximum of 443 wildfires. Figure 1 illustrates the nationwide concentrations of $PM_{2.5}$ and wildfire records in Brazil.

**Association between wildfire PM2.5 and hospital admissions**. Table 3 shows the national average odds ratio of hospital admissions associated with wildfire-related $PM_{2.5}$ in wildfire waves. The national meta-analysis showed that wildfire waves were associated with a 23% (95% confidence interval (CI): 12–33%) increase in respiratory hospital admissions when we accounted for 5 days moving average in Brazil. This was the highest association found for respiratory diseases. For circulatory diseases, the highest risk occurred when we accounted for 2 days moving average, with an estimated increase of 21% (95% CI: 8–35%) in circulatory hospital admissions (Table 3).

**Table 1 Descriptive characteristics of hospital admission events in Brazil, 2008–2018.**

| Health outcome | Subgroup (age and sex) | Number (%) |
|---|---|---|
| Respiratory hospital admissions | | 1,025,454 (50.2) |
| | Sex – Male | 542,767 (52.9) |
| | Sex – Female | 482,687 (47.1) |
| | Age – 0 to 5 years old | 264,061 (25.7) |
| | Age – 35 to 64 years old | 213,074 (20.8) |
| | Age – >64 years old | 283,360 (27.6) |
| Circulatory hospital admissions | | 1,018,584 (49.8) |
| | Sex – Male | 513,943 (50.4) |
| | Sex – Female | 504,641 (49.5) |
| | Age – 0 to 5 years old | 7142 (0.7) |
| | Age – 35 to 64 years old | 474,504 (46.6) |
| | Age – >64 years old | 465,154 (45.7) |

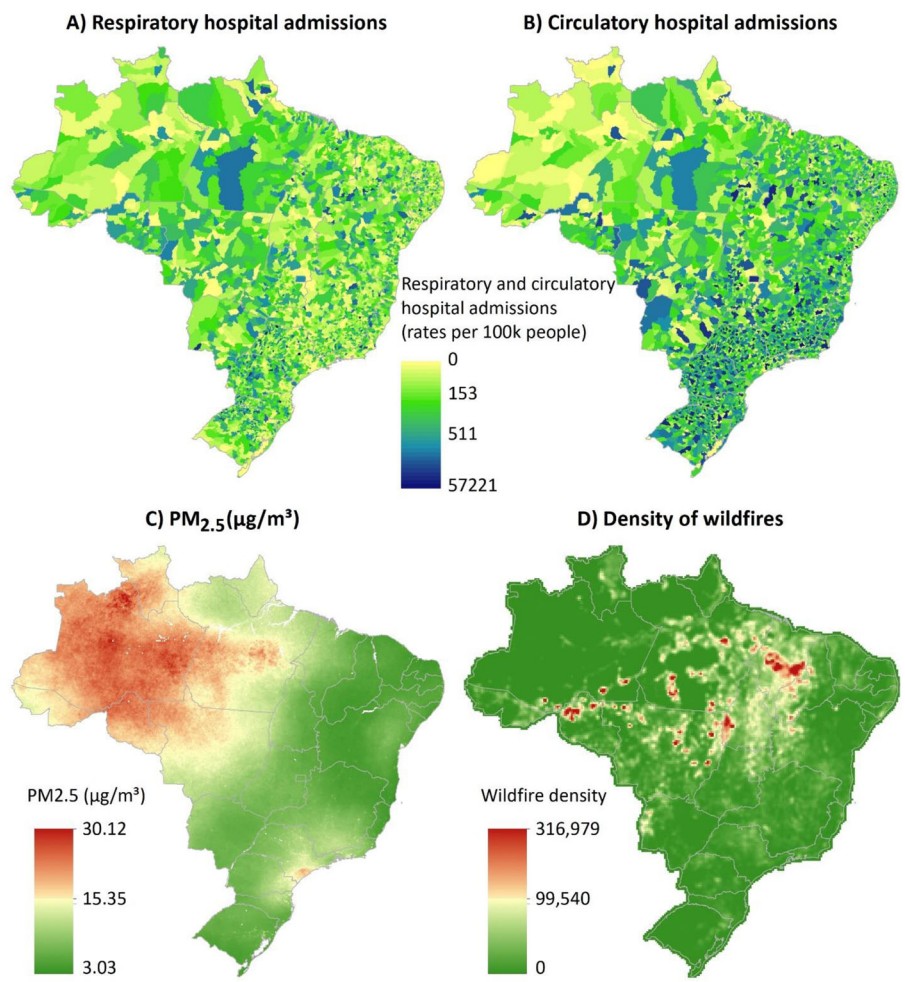

**Fig. 1 Spatial distribution of hospital admissions and exposure variables.** Nationwide occurrences of respiratory and circulatory hospital admissions (total from 2008 to 2018) (**A**, **B**). Concentrations of PM$_{2.5}$ (mean over the study period) (**C**) and wildfire density (**D**) (based on Kernel density with an output cell size of 0.15°; here we accounted for all wildfire records over the study period within a radius of 0.28°) in Brazil.

**Table 2 Summary statistics for wildfire, air pollution, and weather data in Brazil (all the regions), 2008–2018 (the whole study period).**

| Variable | Minimum | First quartile | Mean | Third quartile | Maximum |
|---|---|---|---|---|---|
| Wildfire records | 0 | 1.00 | 4.37 | 4.00 | 443 |
| PM$_{2.5}$ (µg m$^{-3}$) | 0 | 6.42 | 14.69 | 16.375 | 190.08 |
| CO (p.p.b.) | 0 | 95.45 | 154.68 | 177.75 | 201.21 |
| NO$_2$ (p.p.b.) | 0 | 0.75 | 2.58 | 2.75 | 57.75 |
| O$_3$ (p.p.b.) | 0 | 18.40 | 22.77 | 26.75 | 118.45 |
| Temperature (°C) | 2.55 | 22.23 | 24.24 | 26.50 | 34.80 |
| Relative humidity (%) | 21.00 | 71.00 | 78.03 | 87.75 | 100.00 |
| Wind speed (m/s) | 0.23 | 2.17 | 3.16 | 3.92 | 17.02 |
| Wind direction (°) | 1.00 | 85.50 | 135.20 | 175.00 | 358.00 |
| Precipitation (mm/day) | 0 | 0 | 4.12 | 4.00 | 244.00 |

The results of the primary analysis stratified by regions are shown in Fig. 2. In Fig. 3, we present part of our sensitivity analysis, including the subgroup analysis by sex and age. The full results (primary and all sensitivity analysis) are shown in Supplementary Table.

We estimated substantial heterogeneity of the risk of hospital admissions associated with wildfire-related PM$_{2.5}$ in wildfire waves across regions. Overall, North and Midwest were the regions with the highest risk of hospital admissions. We also observed a significant variation of the associations depending on the day of the exposure, assessed by the moving averages. For example, considering the respiratory hospital admissions, in Midwest the highest association was five days following the event (5 days moving average), with an estimated increase of 20% (95% CI: 13–26%) in respiratory disease. For circulatory hospital admissions in Midwest, the highest association was 2 days following the event (2 days moving average), with an estimated increase of 56% (95% CI: 36–76%) in hospital admissions (Fig. 2).

Figure 3 shows the regional percentage increase in the risk of hospital stratified by sex and age. We found in this sensitivity

**Table 3 Odds ratio and 95% CI representing the national average (from meta-analysis) hospital admissions associated with wildfire-related PM$_{2.5}$ in wildfire waves in Brazil (2008–2018) for 1–5 moving averages.**

| Outcome | Moving averages | Odds ratio | Lower 95% CI | Upper 95% CI |
|---|---|---|---|---|
| Respiratory Hospital Admissions | 1 | 1.12 | 1.09 | 1.14 |
| | 2 | 1.12 | 1.07 | 1.17 |
| | 3 | 1.15 | 1.11 | 1.19 |
| | 4 | 1.17 | 1.14 | 1.20 |
| | 5 | 1.23 | 1.12 | 1.33 |
| Circulatory Hospital Admissions | 1 | 1.14 | 1.09 | 1.18 |
| | 2 | 1.21 | 1.08 | 1.35 |
| | 3 | 1.15 | 1.10 | 1.20 |
| | 4 | 1.17 | 1.11 | 1.22 |
| | 5 | 1.20 | 1.12 | 1.29 |

analysis significant groups of ties, which a large number of health events (defined as case day) is out of a large number of subjects (case plus control days). The algorithm in the Survival package may refuse to undertake the task (the computation is infeasible) due to overflow for the subscripts. For most of the regions and moving averages, there is no substantial differences among men and women. For the subgroup analysis by age, overall, our results indicate a higher risk of respiratory hospital admissions for those aged 0–5 years compared with older patients. For circulatory hospital admissions, overall, a higher risk is observed for those aged >64 years.

## Discussion

Here we report significant association of wildfire-related PM$_{2.5}$ in wildfire waves with respiratory and circulatory hospital admissions in Brazil. This is consistent with results from previous Brazilian investigations (limited number of studies, as we presented in the "Introduction"), studies from other countries, and studies of PM$_{2.5}$ in general.

In Brazil, Ignotti et al.[21] estimated the impact on respiratory hospital admissions due to PM$_{2.5}$ emitted from burnings in the Amazon region (for the period 2004–2005) and reported an increase of 8% in child hospitalization and 10% in the hospitalization of the elderly. Carmo et al.[16] reported an increase of 0.29% in respiratory hospital admissions for the children population after the sixth day of exposure in a specific city in the Amazon region. In our study, the North region (where most of the Amazon region is located) had an increase of 21% (95% CI: 7–34%) in respiratory hospital admissions (5 days moving average) for those aged ≤5 years, whereas for those aged ≥65, we estimated an increase of 19% (95% CI: 3.2–35%) at a moving average of 4 days. In the primary analysis, North was identified as the region with the highest risk of hospital admissions due to respiratory diseases, with an estimated increase of 38% (95% CI: 30–47%) in hospital admissions at a moving average of 5 days.

Most of the studies in other countries are focused on the United States and Australia. Some of these investigations include the study in North Carolina—the United States, in which the authors estimated excess relative risk of 66% for asthma at lag day 0 and 42% for heart failure per 100 μg m$^{-3}$ of PM$_{2.5}$[20]. The authors also reported that cardiorespiratory diseases associated with exposure to wildfire smoke are strongly modified by measures of community health and socio-economic factors. This effect modification was also observed in our analyses when we stratified the analysis by Brazilian regions. Another study is in the Western United States, with an estimated 7.2% increase in risk for respiratory hospital admissions during smoke wave days with

high wildfire-specific PM$_{2.5}$ (>37 μg m$^{-3}$); circulatory admissions did not present associations[11].

To our knowledge, few previous studies[14,15,22,23] used the same statistical approach as we used in our study—time-stratified case-crossover study design using conditional logistic regression models. Given the use of the same approach, the comparison of our results with these studies is more reasonable. Johnston et al.[15] found that extreme air pollution events from wildfires in Sydney, Australia, were associated with a 10% increase in cardiovascular mortality. Haikerwal et al.[14] report an increase of 6.98% in the risk of out-of-hospital cardiac arrests associated with an increase in interquartile range of 9.04 μg m$^{-3}$ in PM$_{2.5}$ in Victoria, Australia. In Washington, the United States, Gan et al.[22] found PM$_{2.5}$ from wildfire was associated with an 8% increase in hospital admissions due to asthma. In California, Jones et al.[23] reported that out-of-hospital cardiac arrest risk increased in association with extreme wildfire episodes, with an estimated OR of 1.70 (95% CI: 1.18–2.13) on lag day 2. In our study, in the national meta-analysis, the estimated increase in cardiovascular hospital admissions varied from 14.9% (95% CI: 9–18%) with 1-day moving average to 21% (95% CI: 8–35%) with 2 days moving average. For the national meta-analysis for respiratory diseases, we estimated an increase of 23% (95% CI: 12–33%) in respiratory hospital admissions when we accounted for 5 days moving average. Haikerwal et al.[14] also found a substantial variation of the impact of PM$_{2.5}$ exposure during wildfires on cardiovascular health outcomes in Victoria when they stratified the analyses by age groups and sex. We found the same variation for the subgroup analysis by age but not for the subgroup analysis by sex.

A portion of the regional heterogeneity of the impact of wildfire PM$_{2.5}$ exposure indicated by our results can be explained by the cultural, social, behavioral, and environmental/geographical conditions[10,20,24]. These conditions may determine the use of health services by affected people during wildfire episodes, including (i) the perception and decision of each person to seek medical care after getting symptoms of cardiorespiratory diseases during wildfire events and (ii) the spatial distribution of health care facilities—e.g., hospitals, clinics, and outpatient care centers. For the first one, it is difficult to estimate it in studies with a large population, like our research, given that it relates to individual perceptions and decisions. For the second one, we point out that in Brazil, the distribution of health facilities varies drastically by region. For example, there are many more health facilities per capita in the South than in the Amazon region, including the North. In Brazil, equity in health services is still poorly distributed among the regions. This important condition needs further exploration in studies exploring health impacts from wildfires.

Our results show high concentration of PM$_{2.5}$ in the Amazon region, whereas the regions with high density of wildfire are mostly located in part of the North and Northeast in Brazil. A recent study Wu et al.[25] has shown a mix of chemical species in the air particulates in the Amazon region, including ammonium sulfate (biogenic origin from the rainforest), nitrate (mostly biogenic emissions), elemental carbon (anthropogenic origin), organics, and mineral dust mixed with sea salts (probably during long-range transatlantic transport from Sahara Desert) Wu et al.[25]. In this context, wind speed plays an important role on the atmospheric composition in the Amazon region, which Northeastern wind from the Atlantic Ocean is predominant in the wet season, while in the dry season the predominant wind comes from Southeast (central part of Brazil) Fernandes et al. 2021[26].

Toxicological studies suggest various potential mechanisms via which wildfire-related air pollution might contribute to cardiorespiratory diseases. Wildfires emit or result in a mix of primary and secondary pollutants—particulate pollution (including PM$_{2.5}$ and its elemental and black carbon components), gases (e.g., CO,

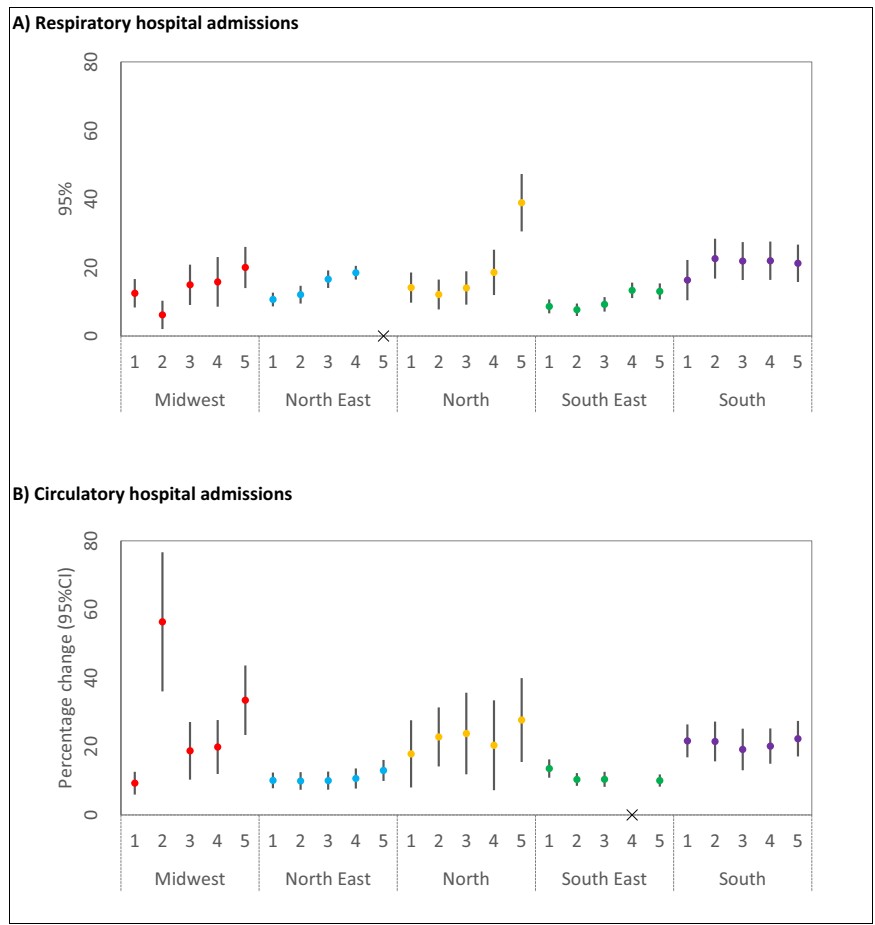

**Fig. 2 Association between hospital admissions and wildfire-related PM2.5.** Regional percentage increase in risk (and 95% CI) of hospital admissions (**A**, respiratory diseases; **B**, circulatory diseases) associated with wildfire-related $PM_{2.5}$ in wildfire waves for the moving averages 1–5 days. Note 1: Numbers in *x*-axis indicate the moving averages. Note 2: "*x*" means that there were very large groups of ties, which a large number of health events (defined as case day) is out of a large number of subjects (case plus control days). The algorithm in the Survival package may refuse to undertake the task (the computation is infeasible) due to overflow for the subscripts. Note 3: Red (Midwest), blue (North East), orange (North), green (South East), and purple (South). Note 4: $n = 2,044,038$ individuals.

$NO_2$, $O_3$) and, depending on what products are burned, there are emissions of other toxic pollutants like benzene or formaldehyde[27]. Evidence has shown that each air pollutant has different toxicologic and physiologic effects on human health[28]. $PM_{2.5}$ has been shown to cause endothelial and vascular dysfunction, oxidative stress, thrombosis, as well as metabolic dysfunction, all of which can contribute to its cardiac effects[28]. For the respiratory system, toxicological studies have found that $PM_{2.5}$ from wildfire induces significant lung toxicity and mutagenic potencies[29], increases neutrophils and protein in lung lavage and by histologic indicators of increased cell influx and edema in the lung[30], and kills lung macrophages by oxidative stress[31].

Our study has several strengths. First, our findings add strength to the evidence that wildfire pollution, such as pollution from other sources, has adverse cardiac and respiratory effects, likely through similar biological mechanisms as shown in toxicological literature. Second, our sample size includes more than 2 million hospital admissions nationwide over 11 years. To our knowledge, this is the study with the largest sample size and the largest study period in Brazil. As we mentioned above, the limited number of previous studies in Brazil have only conducted regional analysis (mostly only in the Amazon region) of the wildfire impact on health over no more than 2 years. Third, we used a modeling approach that indicates the extreme episodes of wildfire, which

we defined as wildfire waves. This allowed us to capture days with elevated concentration, periodic, and short-lived characteristics of wildfire $PM_{2.5}$. Fourth, we accounted for a strong quantitative component in our analyses, looking at the exposure on the day of the health event and the exposure on non-event days with a time-stratified sampling. As described in the "Methods" sections, this reduced the effects of confounding related to the seasonal trend by controlling for time-dependent/independent risk factors.

Our results, however, should be interpreted considering some limitations. First, given the presence of the individual perceptions and decisions to seek medical care after symptoms of cardior-espiratory diseases during wildfire episodes (as mentioned above), there will be different types of persons. For example, there will be people who went to the hospital on the first day of the exposure, people who wait until symptoms become too severe and went to the hospital three days after the exposure, people who got acute symptoms but did not go to the hospitals, etc. Therefore, we probably underestimated the case number and this may affect our results. Second, the definition of wildfire waves may be a source of misclassification, because the number of smoke days will be fixed over a year. Third, the predicted concentration of air pollutants (including the exposure variable $PM_{2.5}$) and the predicted meteorological variables may have resulted in some exposure measurement error. However, we highlight that in our study we were interested in temporal changes of air pollution (daily

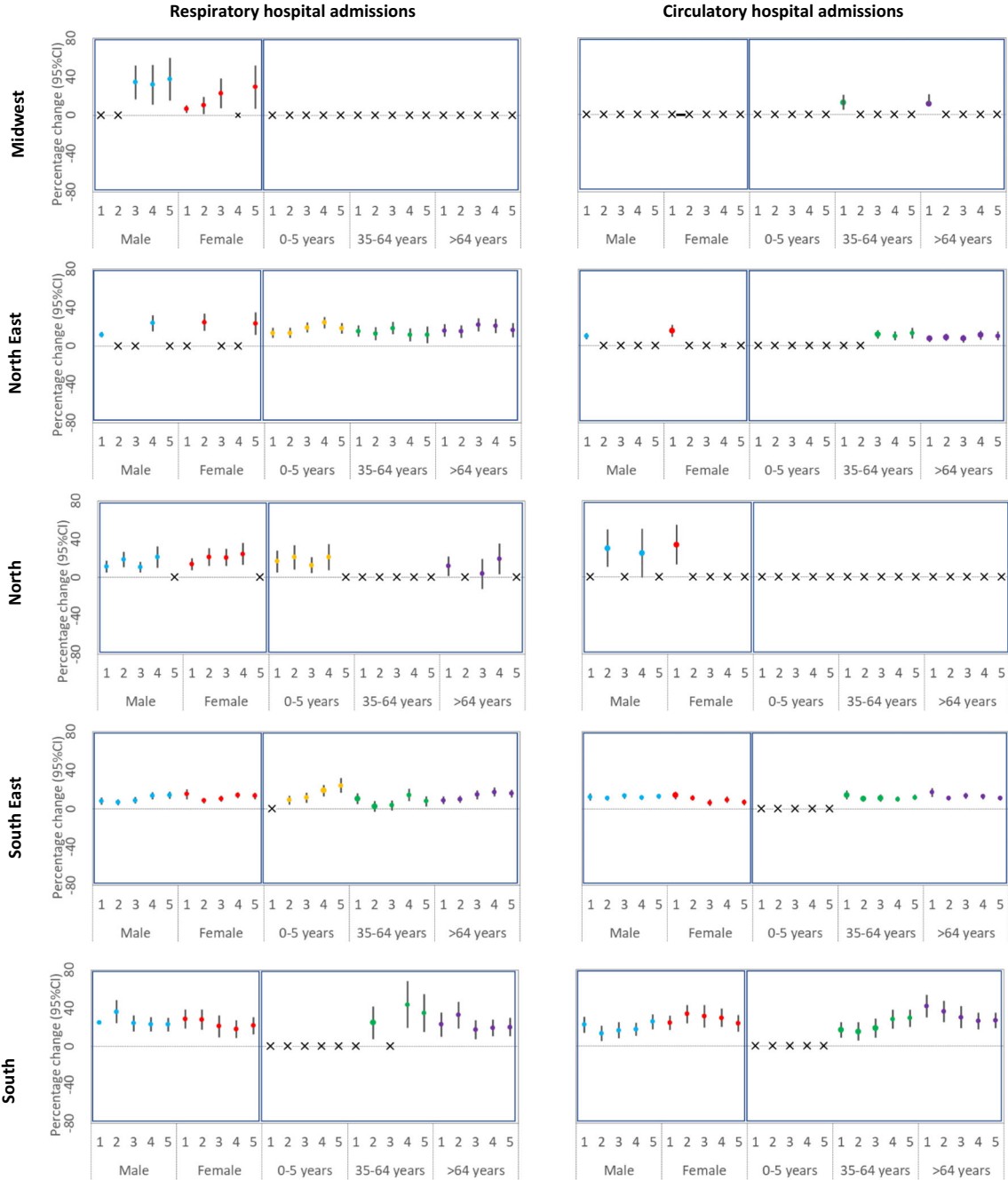

**Fig. 3 Regional percentage increase in risk (and 95% CI) of hospital admissions associated with wildfire-related PM2.5 in wildfire waves for the moving averages 1–5 days by sex and age.** Note 1: *x*-axis indicates the moving averages. Note 2: "*x*" means that there were very large groups of ties, which (a large number of health events (defined as case day) is out of a large number of subjects (case plus control days). The algorithm in the Survival package may refuse to undertake the task (the computation is infeasible) due to overflow for the subscripts. Note 3: Red (Female), blue (Male), orange (0–5 years), green (35–64 years), and purple (>64 years). Norte 4: Left panels (respiratory hospital admissions), right panels (circulatory hospital admissions), panels on the first row (Midwest), panels on the second row (North East), panels on the third row (North), panels on the fourth row (South East), and panels on the last row (South).). Note 5: $n = 2,044,038$ individuals.

variation), and therefore, European Centre for Medium-Range Weather Forecasts (ECMWF) seems to be a reliable source of providing such data as shown by the validation studies mentioned in the "Methods" section. In addition, several other studies have used ECMWF data as source for air pollution and meteorology predictions, including investigations in (i) Portugal on the association between prevailing circulation patterns and particles—$PM_{10}$ and $PM_{2.5}$[32]; (ii) Bavaria, Germany, on the relationship

between weather variables and ambulatory visits due to chronic obstructive pulmonary disease[33]; (iii) Brazil on the potential for a dengue epidemic during the 2014 World Cup[34]; and (iv) Africa on potential predictability of malaria[35]. We highlight that the matching study design controls for individual-level confounding factors as the patients are considered as their controls. Finally, given that the health data is based on filling out of Hospital Admission Authorization Forms (HAAFs), we have to consider a

possibility of errors when the hospital staffs/doctors are filling these forms.

Results from our investigation provide strong epidemiological evidence that air pollution emitted by wildfires is significantly associated with a higher risk of cardiorespiratory hospital admissions, even during small wildfire events. This study can better prepare health experts and environmental scientists by supporting model predictions and public policies according to different levels of fire-prone regions in Brazil.

## Methods

### Exposure data

*Wildfire data.* We accessed wildfire data from the National Institute of Spatial Research of Brazil – Instituto Nacional de Pesquisas Espaciais - INPE (http://queimadas.dgi.inpe.br/queimadas/). The data obtained contain wildfire records, including the date of wildfire occurrence and the geographical location. These data are derived from seven satellite remote sensing observations, including National Ocean and Atmospheric Administration (NOAA)-18, NOAA-19, METOP-B, Moderate Resolution Imaging Spectroradiometer (MODIS) (NASA TERRA and AQUA), VIIRS (NPP-Suomi and NOAA-20), GOES-16, and MSG-3. The INPE process all images from these satellites and then estimates the wildfire occurrences by using a specific satellite as reference. Currently, AQUA is the reference satellite. We accounted for all wildfire records in Brazil based on the reference satellite in the period between 2008 and 2018. Given that each individual in the health data is based on the municipality level (details on the health data are provided in section "Health and population data"), we use Geographic Information System techniques to summarize the number of wildfire occurrences in each Brazilian municipality. There are 5572 municipalities in Brazil, which represent the smallest areas considered by the Brazilian political system. The government groups the municipalities by five regions, including the North, North East, Midwest, South East, and South. In Supplementary Fig. 1, we show the spatial distribution of all municipalities and regions in Brazil.

We defined the concept of "wildfire wave" as any day on which wildfire records and $PM_{2.5}$ concentration exceeded the 99th percentile of the time series from 2008 to 2018 by the Brazilian region (wildfire and $PM_{2.5}$ data, respectively). We used this concept to capture periods with high wildfire occurrences, which allows us to estimate the health effects associated with strong episodes of wildfire-related air pollution. The concept of wildfire wave defined in our study is similar to the concepts of extreme air pollution events from wildfires defined in previous studies[11,15].

*Air pollution data.* Air pollution data were obtained from ensemble models. We accounted for daily $PM_{2.5}$ ($\mu m^{-3}$), CO (p.p.b.), $NO_2$ (p.p.b.), and $O_3$ (p.p.b.) concentrations from 2008 to 2018. The data were accessed from the Environmental Information System for Health (http://queimadas.dgi.inpe.br/queimadas/sisam/v2/dados/download/). This is a database system developed by INPE - National Institute of Spatial Research in Brazil.

The INPE obtained daily concentrations of all these four pollutants from the ECMWF. The ECMWF operates services related to meteorology and air pollution covariates, and implements the Copernicus Atmosphere Monitoring Service (CAMS) on behalf of the European Union including CAMS-Reanalysis and CAMS Near Real Time (CAMS-NRT) forecasts. The CAMS service runs ensemble models using several satellite observations and emission inventories amongst other predictors. We obtained the data at a spatial resolution of 0.125° (~12.5 km) and a temporal resolution of 6 h, including daily estimates for 00, 06, 12, and 18 Universal Time Coordinated. In our analyses, we used CAMS-Reanalysis for the period between 2008 and 2017, and CAMS-NRT for the year 2018.

The validation for the CAMS global model is reported by Inness et al.[36]. Specifically, for the $PM_{2.5}$, the exposure variable in our study, it is evaluated with ground observations of the Aerosol Robotic Network (AERONET). There are over 500 AERONET stations worldwide measuring spectral Aerosol Optical Depth (AOD) with ground-based sun photometers. Among those AERONET stations, about 27 stations are in Brazil. The validation by Inness et al.[36]. estimated a mean bias and SDs from the data provided by the satellite's instruments (included in the CAMS model for aerosols) relative to AERONET data. In South America, the data from satellite's instruments are slightly smaller, with an approximate bias of −0.006 ± 0.128. Another investigation shows that CAMs estimates in South America have a root mean square error (compared with AERONET stations) of 0.268[37]. Other studies have shown that AERONET observation sites in South America has significant representativity for AOD measured by MODIS, aboard TERRA and AQUA satellites[38]. It is noteworthy that MODIS is an instrument included in the CAMS model. This association between AERONET data and AOD from MODIS is significant during the biomass burning seasons in South America, which the $R^2$ (coefficient of determination) for most of the AERONET stations in Brazil was higher than 0.85[38].

We calculated the daily mean temporal resolution for each pollutant. Finally, we aggregated the air pollution data by the municipality, considering the geographical location of the headquarters of each municipality in Brazil.

As we mentioned previously, in this study, the exposure defined as wildfire-related air pollution was based on $PM_{2.5}$ concentration, when it exceeded the 99th percentile of the time series. This cut point is close with the World Health Organization 24-h air quality standard for $PM_{2.5}$ (25 $\mu g\, m^{-3}$). By using this 99% threshold allows our findings to be useful for the public health agencies to act when air pollution standards are higher. As shown in the "Introduction" section, the literature has reported $PM_{2.5}$ as the major pollutant emitted by wildfires. The other pollutants were included in our analyses as control variables. In section "Statistical analyses," we describe the statistical model with all control and confounding variables.

*Weather data.* Meteorological data were retrieved from ensemble models as well, accessed from the ECMWF. Weather data include surface temperature (°C), humidity (%), wind speed (m/s), wind direction (°), and precipitation (mm/day). Temperature, humidity, wind speed, and wind direction were derived from Era-Interim reanalyses, with a spatial resolution of 0.125° and temporal resolution of 6 h. This reanalysis was performed by the ECMWF. Precipitation data was accessed from the Climate Prediction Center and the NOAA. This data has an original spatial resolution of 0.50° (~50 km), with interpolation to 12.5 km, and a temporal resolution of 6 h. As for the air pollution data, we calculated the daily mean temporal resolution for each weather variable and then we aggregated the data by the municipality.

**Health and population data**. The hospital admission data were provided by the Ministry of Health in Brazil. This data was obtained from publicly available database (https://bigdata-metadados.icict.fiocruz.br/dataset/sistema-de-informacoes-hospitalares-do-sus-sihsus) curated by the Ministry of Health in Brazil. The data encompass individual records of hospital admissions in Brazil between 2008 and 2018. Analysis of this data was approved by the Database management group.

Hospital admission information included event date, home municipality, age, sex, race, number of days that patients spend in hospital, and principal diagnosis according to the International Classification of Diseases, version 10 (ICD-10) codes. As discussed in the "Introduction," review studies of the health impacts of wildfire exposure report consistent evidence on the associations between wildfire exposure and cardiorespiratory health effects[10,39]. Therefore, in this study we examined respiratory (ICD-10 codes J00-J99) and cardiovascular (ICD-10 codes I00-I99) diseases. During the period between 2008 and 2018, there were 2,044,038 hospital admissions for cardiorespiratory diseases in Brazil.

**Statistical analyses**. We applied a time-stratified case-crossover study design using conditional logistic regression models. This study design is based on a binary indicator variable for case/control days to compares the exposure (wildfire-related $PM_{2.5}$ in wildfire waves) on the day of the health event (hospital admission; case day) with the exposure on non-event days (control days). We used a time-stratified sampling to select the referent exposure days, which were matched for the day of the week, month, and calendar year of the hospital admission. This allows the comparison of the exposure on the day of a health event on Monday in January in 2008, e.g., with exposures on all other Mondays in January in 2008. It is noteworthy that in this study design, each case period has three or four control periods.

We chose to conduct a matched analysis, because the wildfire-related air pollution exposure is an episodic event. Moreover, the matching approach incorporates some advantages. First, given that the matching periods were close in time, the approach reduces the effects of confounding related to the seasonal trend by controlling for time-dependent risk factors, including the day of the week, season, and long-term trends by matching. Also, people who had the health event were defined as their own controls, allowing for control of all individual-level potential confounders (e.g., socio-economic status, smoking history, pre-existing medical conditions) by design, except for ones that change rapidly.

We used the conditional logistic regression model to estimate the odds ratio (OR) for hospital admissions associated with wildfire-related $PM_{2.5}$ in wildfire waves compared with the background. We adjusted the model for several control/confounding variables, including other air pollutants emitted by wildfires (CO, $NO_2$, and $O_3$), meteorological variables (temperature, relative humidity, precipitation, wind speed, wind direction), topographic variable represented by elevation, spatial terms (latitude, longitude, state, and a binary variable representing the municipalities that are the capitals), and a health variable indicating the number of days that patients spend in hospital.

In the primary analysis, we applied the conditional logistic model described above for each group of health outcomes—respiratory diseases and cardiovascular diseases. We accounted for moving averages for wildfire, air pollutants, and weather variables. We considered five moving averages, including 1-day moving average, 2 days moving average, …, 5 days moving average.

We conducted numerous effect modification and sensitivity analysis by stratifying the analyses by sex, by age (0–5 years old, 35–64 years old, and >64 years old), by excluding the control pollutants from the model (CO, $NO_2$, and $O_3$), by excluding race, by excluding some spatial terms (state and latitude/longitude), and by accounting for cardiorespiratory hospital admissions (cardiovascular and respiratory hospital admissions together). We applied the moving average for each one of these stratifications (subgroup analysis). All statistical analyses were performed in R, using the statistical package Survival (clogit function).

The primary analysis and all the sensitivity analyses were conducted individually for each one of the five Brazilian regions (Appendix 1 shows the spatial distribution of these regions). We performed this subgroup analysis by region to capture the regional heterogeneity of landscape in Brazil (e.g., Amazon Forest, Atlantic Forest, Pantanal, etc.), which is strongly correlated with wildfire occurrences. Then, region-specific OR were meta-analyzed to estimate national average hospital admissions associated with wildfire-related $PM_{2.5}$ in wildfire waves. We accounted for intra- and inter-region variability by applying regression meta-analysis with random effects.

**Reporting summary**. Further information on research design is available in the Nature Research Reporting Summary linked to this article.

## Data availability

All data supporting the findings described in this manuscript are available in the article and in the Supplementary Information, and from the corresponding author upon reasonable request. The health data used in this study are available in the Ministry of Health in Brazil (https://bigdata-metadados.icict.fiocruz.br/dataset/sistema-de-informacoes-hospitalares-do-sus-sihsus) or the database can be directly downloaded from https://bigdata-arquivos.icict.fiocruz.br/SIH/ETLSIH.zip). The wildfire data used in this study are available in the INPE database under this: link http://queimadas.dgi.inpe.br/queimadas/bdqueimadas. The air pollution and weather data used in this study are available in the INPE database under this following link: http://queimadas.dgi.inpe.br/queimadas/sisam/v2/dados/download/.

## Code availabilitty

The code that supports the findings of this study are available under this link: https://github.com/weeberb/Wildfire-and-Hospital-Admissions-in-Brazil.git.

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

## Acknowledgements

This work was supported by the Brazilian Agencies National Council for Scientific and Technological Development (CNPq) and by the Ministry of Science, Technology and Innovation (MCTI). HA is supported by Novo Nordisk Foundation Challenge Programme: Harnessing the Power of Big Data to Address the Societal Challenge of Aging (NNF17OC0027812). JS is supported by U.S. EPA grant RD-83587201–0. The contents of this publication are solely the responsibility of the grantee and do not necessarily represent the official views of the US EPA. Further, the US EPA does not endorse the purchase of any commercial products or services mentioned in the publication.

## Author contributions

W.R. initiated the study, synthesized data, and performed the analysis. R.M., H.A. and J.S. contributed to the model development. W.R., R.M., H.A., J.S. and D.G. interpreted the results. All authors have read and approved the manuscript for publication.

## Competing interests

The authors declare no competing interests.
