## [Peer Review File · Nature Communications]

Reviewers' Comments:

Reviewer #1:

Remarks to the Author:

Thank you for the opportunity to read and review the manuscript titled "Health impacts of wildfire-related air pollution in Brazil: A nationwide study of more than 1.2 million hospital admissions between 2008 and 2018". This study examined the effects of wildfire smoke on cardiopulmonary outcomes in population that is often underrepresented in environmental epidemiology. I think the paper is very well written, analytical methods seem appropriate and the results are well interpreted. I only have two major comments and a few minor ones.

- Use of Copernicus model for exposure is concerning to me. I am not very familiar with the performance or construction of the model but I remember that the model caused headline news for sending wrong air quality predictions to the public in North America. I believe one of the weather apps used Copernicus data. This concern is not lessened by the manuscript not providing effort to validate the model.
- Definition of wildfire wave is troubling to me in two ways. Using a 99% threshold procedure requires an analyst to investigate the data before identifying the treatment. Such treatment variables are usually not permitted and although the use is intuitive it creates a significant challenge to the interpretation of results.
- "x means that there were very large groups of ties (a large number of events out of a large number of subjects). The algorithm in 306 the Survival package may refuse to undertake the task (the computation is infeasible) due to overflow for the subscripts." – I could not make sense of this sentence so I don't know how to interpret the results.

Figure 1- Please convert hospitalizations to rates per 100K people. Absolute counts just give us sense of where people are not very the vulnerabilities are. Additionally can the authors provide some explanation why pm is so high in the western amazon where there are no fires.

Line 259 – Please clarify what the summary statistics are reported in the table. Are these statistics referring to daily or by region?

Line 292 We found in this sensitivity analysis significant groups of ties (many events out of many 292 subjects). The algorithm in the Survival package may refuse to undertake the task (the computation 293 is infeasible) due to overflow for the subscripts.

Line 335 "0.6% for asthma at lag day 0, and 0.42% for heart failure - these estimates were reported per 100 µg/m³ of PM_{2.5};" referring to Rappold 2012. I don't remember exactly the risk estimates from this paper but I do remember that Rappold 2011 and 2012 addressed the same fire episode and that the notable the first paper was first to report notable cardiovascular effects which were on the order of 40% increase in risk per day of exposure not 0.4%. This would be good to double check.

Line 344 "To our knowledge, there are two previous studies (Haikerwal et al. 2015; Johnston et al. 2011) that used the same statistical approach as we used in our study - time-stratified case-crossover study design using conditional logistic regression models" -- I believe there were few other studies that used case crossover eg Liu et al, Gan et al, and Jones et al and maybe the San Diego fires paper.

Line 360 Add Reid 2018 (or 2017)

Reviewer #2:

Remarks to the Author:

The article is well written and addresses an important and little studied topic in Brazil. Therefore, I recommend publication after corrections.

General remarks

Separate the unit from the number – e.g. 10 %

µg m⁻³ – correct form

Line 125. "We accounted for all wildfire records in Brazil captured by these satellites in the period between 2008 and 2018." Using the data from all satellites, is there not an overlap in the number

of fires?

Air pollution data - what is the error associated with satellite measurements? Have concentrations been validated with ground-based observational data?

The North region, where hospitalizations predominated, is a region that uses firewood and rudimentary stoves. Wouldn't this be one of the causes of hospitalizations as well?

Discuss the limitations of and the reliability of the DATASUS data used.

RESPONSES TO REVIEWER #1:

Thank you for the opportunity to read and review the manuscript titled “Health impacts of wildfire-related air pollution in Brazil: A nationwide study of more than 1 2 million hospital admissions between 2008 and 2018”. This study examined the effects of wildfire smoke on cardiopulmonary outcomes in population that is often underrepresented in environmental epidemiology. I think the paper is very well written, analytical methods seem appropriate and the results are well interpreted. I only have two major comments and a few minor ones.

Response:

We thank the reviewer for acknowledging the worth of the present manuscript. The questions and comments are taken into account in the revised manuscript. Detailed responses to each question are given below.

Use of Copernicus model for exposure is concerning to me. I am not very familiar with the performance or construction of the model but I remember that the model caused headline news for sending wrong air quality predictions to the public in North America. I believe one of the weather apps used Copernicus data. This concern is not lessened by the manuscript not providing effort to validate the model.

Response:

Thank you for the comment. We have addressed this issue in the methods section. First, in line 151-160, we clarified the information on data source. This is the text edited in the manuscript:

“The INPE obtained daily concentrations of all these four pollutants from the European Centre for Medium-Range Weather Forecasts (ECMWF). The ECMWF operates services related to meteorology and air pollution covariates, and implements the Copernicus Atmosphere Monitoring Service (CAMS) on behalf of the European Union including CAMS-Reanalysis and CAMS Near Real Time (CAMS-NRT) forecasts. The CAMS service runs ensemble models using several satellite observations and emission inventories amongst other predictors. We obtained the data at a spatial resolution of 0.125 degrees (approximately 12.5 km) and a temporal resolution of 6 hours, including daily estimates for 00, 06, 12, and 18 UTC (Universal Time Coordinated). In our analyses, we used CAMS-Reanalysis for the period between 2008- 2017 and CAMS-NRT for the year 2018.”

Then, in line 169-185, we have addressed the issue related to the validation of CAMS estimates. This is the text included in the manuscript:

“The validation for the CAMS global model is reported by Inness et al. (2018). Specifically, for the $PM_{2.5}$, the exposure variable in our study, it is evaluated with ground observations of the Aerosol Robotic Network (AERONET). There are about 400 AERONET stations worldwide measuring spectral Aerosol Optical Depth (AOD) with ground-based sun photometers. Among those AERONET stations, about 27 stations are in Brazil. The validation by Inness et al. (2018) estimated a mean bias and standard deviations from the data provided by the satellite’s instruments (included in the CAMS model for aerosols) relative to AERONET data. In South America, the data from satellite’s instruments are slightly smaller, with an approximate bias of -0.006 ± 0.128 . Another investigation shows that CAMS estimates in South America have a root mean square error – RMSE (compared with AERONET stations) of 0.268 (Gueymard and Yang 2020). Other studies have shown that AERONET observation sites in South America has significant representativity for AOD measured by Moderate Resolution Imaging Spectroradiometer (MODIS), aboard TERRA and AQUA satellites (Hoelzemann et al. 2009). Note that MODIS is an instrument included in the CAMS model. This association between AERONET data and AOD from MODIS is significant during the biomass burning seasons in South America, which the R^2 (coefficient of determination) for most of the AERONET stations in Brazil was higher than 0.85 (Hoelzemann et al. 2009).”

Also, in the discussion section, we cited some studies that have used ECMWF air pollution and meteorology predictions. This is the text included in the discussion section (line 457-469):

“Second, the predicted concentration of air pollutants (including the exposure variable $PM_{2.5}$) and the predicted meteorological variables may have resulted in some exposure measurement error. However, we highlight that in our study we were interested in temporal changes of air pollution (daily variation), and therefore, ECMWF seems to be a reliable source of providing such data as shown by the validation studies mentioned in the methods section. In addition, several other studies have used ECMWF data as source for air pollution and meteorology predictions, including investigations in i) Portugal on the association between prevailing circulation patterns and particles - PM_{10} and $PM_{2.5}$ (Cavaleiro et al. 2021), ii) Bavaria, Germany, on the relationship

between weather variables and ambulatory visits due to chronic obstructive pulmonary disease (Ferrari et al. 2012), iii) Brazil on the potential for a dengue epidemic during the 2014 World Cup (Lowe et al. 2014), and iv) Africa on potential predictability of malaria (Tompkins and di Giuseppe 2015).”

Finally, we are not aware of the issue mentioned by the reviewer “I remember that the model caused headline news for sending wrong air quality predictions to the public in North America. I believe one of the weather apps used Copernicus data”. We have tried to find which app was that and what was the news. We were unable to find the news, and for the app, we found one named as "Windy" app. We are not sure if this is the app mentioned by the reviewer. It seems that this “Windy” app is not a good source of scientific work and judgment.

Definition of wildfire wave is troubling to me in two ways. Using a 99% threshold procedure requires an analyst to investigate the data before identifying the treatment. Such treatment variables are usually not permitted and although the use is intuitive it creates a significant challenge to the interpretation of results.

Response:

Thank you for the feedback. We believe that the approach used to define wildfire wave is reliable for episodic events, such as wildfire-related air pollution exposure. As we mentioned in the discussion section (line 444-446), this modeling approach (defined as “wildfire waves”) indicates the extreme episodes of wildfire. This allowed us to capture days with elevated concentration, periodic, and short-lived characteristics of wildfire PM_{2.5}. We believe that this approach incorporates advantages by integrating it in the time-stratified case-crossover study design using conditional logistic regression models.

We also highlight that two previous important studies used similar approach (we mentioned this in line 142-143). One study by Johnston et al. (2011), which the authors estimated the impact of extreme air pollution events from wildfire on mortality in Sydney, Australia. The second study by Liu et al. (2017) that estimates wildfire-specific PM_{2.5} and risk of hospital admissions in the Western US.

Finally, the 99th percentile as the cut point corresponds closely with the World Health Organization 24-h air quality standard for PM_{2.5} (25 µg/m³). This allows our results to be useful for the public health authorities that are required to act when air quality standards are violated. We have included this justification in the manuscript (line 191-193):

“This cut point is close with the World Health Organization 24-h air quality standard for PM_{2.5} (25 µg/m³). By using this 99% threshold allows our findings to be useful for the public health agencies to act when air pollution standards are higher.”

“x means that there were very large groups of ties (a large number of events out of a large number of subjects). The algorithm in the Survival package may refuse to undertake the task (the computation is infeasible) due to overflow for the subscripts. “– I could not make sense of this sentence so I don’t know how to interpret the results.

Response:

Thank you for the comment. According to the documentation of the Survival package in R, “If a particular strata had say 10 events out of 20 subjects we have to add up a denominator that involves all possible ways of choosing 10 out of 20, which is $20!/(10! 10!) = 184756$ terms. Gail et al. (1980) describe a fast recursion method which partly ameliorates this; it was incorporated into version 2.36-11 of the survival package. The computation remains infeasible for very large groups of ties, say 100 ties out of 500 subjects, and may even lead to integer overflow for the subscripts – in this latter case the routine will refuse to undertake the task”. In our case, events represent the hospital admissions – the case day. Subjects are the cases plus control days. We have clarified this information in the manuscript. Now, the note 2 for Figure 2 and 3 says:

“x” means that there were very large groups of ties, which a large number of health events (defined as case day) is out of a large number of subjects (case plus control days). The algorithm in the Survival package may refuse to undertake the task (the computation is infeasible) due to overflow for the subscripts.”

Figure 1- Please convert hospitalizations to rates per 100K people. Absolute counts just give us sense of where people are not very the vulnerabilities are. Additionally can the authors provide some explanation why pm is so high in the western amazon where there are no fires.

Response:

Thank you for the suggestion. We have converted hospitalizations to rates in Figure 1. Please see below the new Figure. Regarding the elevated concentration of PM_{2.5} in the Western Amazon, we have included the following explanation in the discussion section (line 414-423).

“Our results show high concentration of PM_{2.5} in the Amazon region, while the regions with high density of wildfire are mostly located in part of the North and Northeast in Brazil. A recent study (Wu et al. 2019) has shown a mix of chemical species in the air

particulates in the Amazon region, including ammonium sulfate (biogenic origin from the rainforest), nitrate (mostly biogenic emissions), elemental carbon (anthropogenic origin), organics, mineral dust mixed with sea salts (probably during long-range transatlantic transport from Sahara Desert) (Wu et al. 2019). In this context, wind speed plays an important role on the atmospheric composition in the Amazon region, which Northeastern wind from the Atlantic Ocean is predominant in the wet season, while in the dry season the predominant wind comes from Southeast (central part of Brazil) (Fernandes et al. 2021).”

Line 259 – Please clarify what the summary statistics are reported in the table. Are these statistics referring to daily or by region?

Response:

Thank you for the comment. This table shows summary statistics for the whole study area and study period. We have clarified this information. This is the new title of this figure: “Table 2 – Summary

statistics for wildfire, air pollution, and weather data in Brazil (all the regions), 2008-2018 (the whole study period).”

Line 292 We found in this sensitivity analysis significant groups of ties (many events out of many subjects). The algorithm in the Survival package may refuse to undertake the task (the computation is infeasible) due to overflow for the subscripts.

Response:

Thank you for the comment. We have clarified this information in the manuscript. This is the new sentence (line 325-328):

“We found in this sensitivity analysis significant groups of ties, which a large number of health events (defined as case day) is out of a large number of subjects (case plus control days).. The algorithm in the Survival package may refuse to undertake the task (the computation is infeasible) due to overflow for the subscripts.”

Line 335 “0.6% for asthma at lag day 0, and 0.42% for heart failure - these estimates were reported per 100 $\mu\text{g}/\text{m}^3$ of $\text{PM}_{2.5}$,” referring to Rappold 2012. I don’t remember exactly the risk estimates from this paper but I do remember that Rappold 2011 and 2012 addressed the same fire episode and that the notable the first paper was first to report notable cardiovascular effects which were on the order of 40% increase in risk per day of exposure not 0.4%. This would be good to double check.

Response:

We appreciate the reviewer’s comment. It is correct, Rappold et al. (2012) reported an increase of 40%. As we had mentioned in the manuscript, we had converted the coefficient for the original slope (0.4%), because the Rappold et al. (2012) reported the estimates per 100 $\mu\text{g}/\text{m}^3$ of $\text{PM}_{2.5}$. To avoid any confusion, we have decided to show their results as the exactly way they reported in the paper. This is the new sentence (line 371-374):

“... in which the authors estimated excess relative risk of 66% for asthma at lag day 0, and 42% for heart failure per 100 $\mu\text{g}/\text{m}^3$ of $\text{PM}_{2.5}$ (Rappold et al. 2012).”

Line 344 “To our knowledge, there are two previous studies (Haikerwal et al. 2015; Johnston et al. 2011) that used the same statistical approach as we used in our study - time-stratified case-crossover study design using conditional logistic regression models” -- I believe there were few other studies that used case crossover eg Liu et al, Gan et al, and Jones et al and maybe the San Diego fires paper.

Response:

Thank you for this feedback. We have edited this paragraph in the discussion section (line 381-397):

“To our knowledge, few previous studies (Gan et al. 2017; Haikerwal et al. 2015; Johnston et al. 2011; Jones et al. 2020) used the same statistical approach as we used in our study - time-stratified case-crossover study design using conditional logistic regression models. Given the use of the same approach, the comparison of our results with these studies is more reasonable. Johnston et al. (2011) found that extreme air pollution events from wildfires in Sydney, Australia, were associated with a 10% increase in cardiovascular mortality. Haikerwal et al. (2015) report an increase of 6.98% in the risk of out-of-hospital cardiac arrests associated with an increase in interquartile range of 9.04 $\mu\text{g}/\text{m}^3$ in $\text{PM}_{2.5}$ in Victoria, Australia. In Washington, the US, Gan et al. (2017) found $\text{PM}_{2.5}$ from wildfire was associated with an 8% increase in hospital admissions due to asthma. In California, Jones et al. (2020) reported that out-of-hospital cardiac arrest risk increased in association with extreme wildfire episodes, with an estimated OR of 1.70 (95%CI: 1.18-2.13) on lag day 2. In our study, in the national meta-analysis, the estimated increase in cardiovascular hospital admissions varied from 14.9% (95%CI: 9% - 18%) with 1-day moving average to 21% (95%CI: 8% - 35%) with 2-days moving average. For the national meta-analysis for respiratory diseases, we estimated an increase of 23% (95%CI: 12% - 33%) in respiratory hospital admissions when we accounted for 5-days moving average”.

As you can see, we have added Gan et al and Jones et al. We checked Liu et al and we found that they used a different approach – they used generalized linear mixed model. For the last suggestion, San Diego fires paper, we did not find this paper. If the reviewer shares the full reference of this paper, we will be happy to cite it in our manuscript.

Line 360 Add Reid 2018 (or 2017)

Response:

Thank you for the suggestion. We have added Reid.

RESPONSES TO REVIEWER #2:

The article is well written and addresses an important and little studied topic in Brazil. Therefore, I recommend publication after corrections.

Response:

We thank the reviewer for the careful reading of the manuscript and the associated questions and feedback that were provided. We also thank the reviewer for acknowledging the worth of the present manuscript. Detailed responses to each question are given below.

General remarks

Separate the unit from the number – e.g. 10 %

Response:

Thank you for the comment. We have fixed that.

$\mu\text{g m}^{-3}$ – correct form

Response:

Thanks for the comment. We have corrected this.

Line 125. “We accounted for all wildfire records in Brazil captured by these satellites in the period between 2008 and 2018.” Using the data from all satellites, is there not an overlap in the number of fires?

Response:

Thank you for this important question. We have clarified this sentence. This is the new sentence in the manuscript (line 126-129):

“The INPE process all images from these satellites and then estimates the wildfire occurrences by using a specific satellite as reference. Currently, AQUA is the reference satellite. We accounted for all wildfire records in Brazil based on the reference satellite in the period between 2008 and 2018.”

Air pollution data - what is the error associated with satellite measurements? Have concentrations been validated with ground-based observational data?

Response:

Thank you for the comment. Reviewer #1 above raised a similar question. We pasted below part of the response that we wrote for reviewer #1. We believe that this response will answer your question:

We have addressed this issue in the methods section. First, in line 151-160, we clarified the information on data source. This is the text edited in the manuscript:

“The INPE obtained daily concentrations of all these four pollutants from the European Centre for Medium-Range Weather Forecasts (ECMWF). The ECMWF operates services related to meteorology and air pollution covariates, and implements the Copernicus Atmosphere Monitoring Service (CAMS) on behalf of the European Union including CAMS-Reanalysis and CAMS Near Real Time (CAMS-NRT) forecasts. The CAMS service runs ensemble models using several satellite observations and emission inventories amongst other predictors. We obtained the data at a spatial resolution of 0.125 degrees (approximately 12.5 km) and a temporal resolution of 6 hours, including daily estimates for 00, 06, 12, and 18 UTC (Universal Time Coordinated). In our analyses, we used CAMS-Reanalysis for the period between 2008- 2017 and CAMS-NRT for the year 2018..”

Then, in line 169-185, we have addressed the issue related to the validation of CAMS estimates. This is the text included in the manuscript:

“The validation for the CAMS global model is reported by Inness et al. (2018). Specifically, for the PM_{2.5}, the exposure variable in our study, it is evaluated with ground observations of the Aerosol Robotic Network (AERONET). There are about 400 AERONET stations worldwide measuring spectral Aerosol Optical Depth (AOD) with ground-based sun photometers. Among those AERONET stations, about 27 stations are in Brazil. The validation by Inness et al. (2018) estimated a mean bias and standard deviations from the data provided by the satellite’s instruments (included in the CAMS model for aerosols) relative to AERONET data. In South America, the data from satellite’s instruments are slightly smaller, with an approximate bias of -0.006 ± 0.128 . Another investigation shows that CAMS estimates in South America have a root mean square error – RMSE (compared with AERONET stations) of 0.268 (Gueymard and Yang 2020). Other studies have shown that AERONET observation sites in South America has significant representativity for AOD measured by Moderate Resolution Imaging Spectroradiometer (MODIS), aboard TERRA and AQUA satellites (Hoelzemann et al. 2009). Note that MODIS is an instrument included in the CAMS model. This association between AERONET data and AOD from MODIS is significant

during the biomass burning seasons in South America, which the R^2 (coefficient of determination) for most of the AERONET stations in Brazil was higher than 0.85 (Hoelzemann et al. 2009).”

Also, in the discussion section, we cited some studies that have used ECMWF air pollution and meteorology predictions. This is the text included in the discussion section (line 457-469):

“Second, the location the predicted concentration of air pollutants (including the exposure variable $PM_{2.5}$) and the predicted meteorological variables may have resulted in some exposure measurement error. However, we highlight that in our study we were interested in temporal changes of air pollution (daily variation), and therefore, ECMWF seems to be a reliable source of providing such data as shown by the validation studies mentioned in the methods section. In addition, several other studies have used ECMWF data as source for air pollution and meteorology predictions, including investigations in i) Portugal on the association between prevailing circulation patterns and particles - PM_{10} and $PM_{2.5}$ (Cavaleiro et al. 2021), ii) Bavaria, Germany, on the relationship between weather variables and ambulatory visits due to chronic obstructive pulmonary disease (Ferrari et al. 2012), iii) Brazil on the potential for a dengue epidemic during the 2014 World Cup (Lowe et al. 2014), and iv) Africa on potential predictability of malaria (Tompkins and di Giuseppe 2015).”

The North region, where hospitalizations predominated, is a region that uses firewood and rudimentary stoves. Wouldn't this be one of the causes of hospitalizations as well?

Response:

Thank you for raising this issue. We agree with the reviewer, some other air pollution sources (e.g., firewood and rudimentary stoves) may be contributing to a significant number of hospitalizations. But the problem is that we do not have a credible source of data (e.g., official statistics, scientific study-based surveys) estimating the number of people exposed to this indoor air pollution source in Brazil, including the North. We do not know how many people are using these rudimentary stoves and for those exposed populations, we do not have an epidemiological study in Brazil reporting the health effects.

Another issue is that we edited Figure 1. Now we illustrate the hospitalizations as a proportion (rates per 100k people). We can see now that the municipalities with high hospitalizations are very heterogeneous over Brazil, especially for circulatory hospital admissions. Below is the new Figure 1.

Discuss the limitations of and the reliability of the DATASUS data used.

Response:

Thank you for this suggestion. We have added the following sentence in the discussion section (line 470-472).

“Finally, given that the health data is based on filling out of Hospital Admission Authorization Forms, we have to consider a possibility of errors when the hospital staffs/doctors are filling these forms”.

Reviewers' Comments:

Reviewer #1:

Remarks to the Author:

Thank you for making an earnest effort to address comments. It is my preference that wildfire is defined prior to wildfire not after as a basic principle of causal inference rather than to add citations of how others made the same poor definition. I also understand that the definition was put together because of lack of a better choice (personally communication with both authors that were cited) so its better to acknowledge this rather than to propagate.

Reviewer #2:

Remarks to the Author:

The manuscript was well reviewed and can be accepted for publication.

RESPONSES TO REVIEWER #1:

Thank you for making an earnest effort to address comments. It is my preference that wildfire is defined prior to wildfire not after as a basic principle of causal inference rather than to add citations of how others made the same poor definition. I also understand that the definition was put together because of lack of a better choice (personally communication with both authors that were cited) so its better to acknowledge this rather than to propagate.

Response:

We are grateful for your consideration of this manuscript. We have addressed this comment by editing the last paragraph in section 2.1.1 (lines 135-143). Now, we first define “wildfire wave”. Then we explain the concept. Finally, we cite the references. We believe that we have addressed this remaining concern. This is the edited paragraph.

“We defined the concept of “wildfire wave” as any day on which wildfire records and $PM_{2.5}$ concentration exceeded the 99th percentile of the time series by the Brazilian region (wildfire and $PM_{2.5}$ data, respectively). We used this concept to capture periods with high wildfire occurrences, which allows us to estimate the health effects associated with strong episodes of wildfire-related air pollution. The concept of wildfire wave defined in our study is similar to the concepts of extreme air pollution events from wildfires defined in previous studies (Johnston et al. 2011; Liu et al. 2017).”

RESPONSES TO REVIEWER #2:

The manuscript was well reviewed and can be accepted for publication.

Response:

We thank the reviewer for acknowledging the worth of the present manuscript.